# Non-invasive real-time genomic monitoring of the critically endangered kākāpō

Lara Urban[1,2,3,4]*, Allison K Miller[1], Daryl Eason[5], Deidre Vercoe[5], Megan Shaffer[6], Shaun P Wilkinson[6], Gert-Jan Jeunen[1], Neil J Gemmell[1†], Andrew Digby[5†]

[1]Department of Anatomy, University of Otago, Dunedin, New Zealand; [2]Helmholtz Pioneer Campus, Helmholtz Zentrum Muenchen, Neuherberg, Germany; [3]Helmholtz AI, Helmholtz Zentrum Muenchen, Neuherberg, Germany; [4]Technical University of Munich, School of Life Sciences, Freising, Germany; [5]Kākāpō Recovery Programme, Department of Conservation, Invercargill, New Zealand; [6]Wilderlab NZ Ltd, Wellington, New Zealand

**Abstract** We used non-invasive real-time genomic approaches to monitor one of the last surviving populations of the critically endangered kākāpō (*Strigops habroptilus*). We first established an environmental DNA metabarcoding protocol to identify the distribution of kākāpō and other vertebrate species in a highly localized manner using soil samples. Harnessing real-time nanopore sequencing and the high-quality kākāpō reference genome, we then extracted species-specific DNA from soil. We combined long read-based haplotype phasing with known individual genomic variation in the kākāpō population to identify the presence of individuals, and confirmed these genomically informed predictions through detailed metadata on kākāpō distributions. This study shows that individual identification is feasible through nanopore sequencing of environmental DNA, with important implications for future efforts in the application of genomics to the conservation of rare species, potentially expanding the application of real-time environmental DNA research from monitoring species distribution to inferring fitness parameters such as genomic diversity and inbreeding.

*For correspondence:
lara.h.urban@gmail.com

†These authors contributed equally to this work

## eLife assessment

This work presents **important** findings regarding the use of soil environmental DNA for non-invasive monitoring of the endangered kākāpō parrot population in New Zealand. The approach based on sequence analysis is **convincing** but comparisons to established methods are lacking. The tools presented in this study are innovative and will be relevant to those working with environmental DNA and the conservation of biodiversity.

## Introduction

Across the world, wild populations are declining at an alarming rate (*Ceballos et al., 2017*). The consequent small population sizes directly increase the risk of species extinction and result in a loss of genomic diversity (*Charlesworth and Charlesworth, 1987*), which further impairs resilience to environmental fluctuations (*Frankham, 2005*). Rapidly assessing population fluctuations by monitoring individuals and their genomic diversity is therefore a key tool for modern conservation programs of critically endangered species. Obtaining such data however usually requires the capture and handling of the target species, such as transmitter fitting for individual tracing or blood sampling for genomic analysis. Non-invasive monitoring of individuals and their genomic diversity

based on hair, feathers, or fecal samples has successfully been applied to endangered populations (*Khan et al., 2020*; *Ramón-Laca et al., 2018*), reducing costs as well as disturbance, stress and risk of injury in wild species. We are, however, still in search of a step change that would allow genomic data to be obtained directly from environmental samples such as soil and water, which are easily and universally accessible for any species around the world. Here, we report a significant contribution to this step change by combining environmental DNA (eDNA) approaches with real-time sequencing enabled by nanopore sequencing to analyze species-specific genomic data from environmental material.

The analysis of eDNA, DNA fragments isolated from environmental sources such as water, soil or, most recently, air (*Clare et al., 2021*), has significantly advanced conservation biology and biodiversity management by informing about species presence and variety (*Ruppert et al., 2019*). Most eDNA research relies on metabarcoding to identify species compositions in aquatic and terrestrial ecosystems (*Thomsen and Willerslev, 2015*). To date, eDNA studies have assessed the accuracy of species detection (*Jeunen et al., 2020*; *Murakami et al., 2019*) and quantification (*Sassoubre et al., 2016*; *Uthicke et al., 2018*), and have even been employed directly in the field (*Truelove et al., 2019*; *Urban et al., 2021*). Many eDNA studies focus on water as the source of DNA due to relatively straightforward processing through filtering (*Ushio et al., 2018*). The application of soil eDNA, on the other hand, has evolved from studying fungal and bacterial diversity (*Delmont et al., 2011*; *Edwards and Zak, 2010*) to the analysis of a wide range of taxa of past and present ecosystems (*Edwards et al., 2018*; *Epp et al., 2012*; *Foucher et al., 2020*; *Rota et al., 2020*), specifically of endangered species (*Walker et al., 2017*; *Kucherenko et al., 2018*; *Leempoel et al., 2020*).

While traditional eDNA analysis can discover the presence and distribution of species, information about a species' characteristics such as its population structure or genomic diversity have rarely been retrieved from environmental samples beyond mitochondrial diversity (*Barnes and Turner, 2016*; *Sigsgaard et al., 2020*). Previous studies identified nuclear microsatellites to discern individuals, including research on snow footprints (*Hellström et al., 2019*), phylogenetic inferences in the silver carp (*Stepien et al., 2019*), and comparisons between eDNA- and tissue-derived allele frequencies in the round goby (*Andres et al., 2021*). Shotgun sequencing of ancient DNA from cave sediments has further enabled the creation of the environmental genome of extinct species, potentially expanding ancient eDNA research into the population genomics domain (*Gelabert et al., 2021*; *Pedersen et al., 2021*; *Zavala et al., 2021*). More recently, *Farrell et al., 2022* have been the first to showcase the potential to unlock information about individual and population-level diversity via shotgun sequencing of DNA extracted from sand to infer individual turtle source populations (*Farrell et al., 2022*).

Here, we use non-invasive real-time genomics to monitor one of the last surviving populations of the critically endangered kākāpō (*Strigops habroptilus*). The kākāpō (*Strigops habroptilus*) is a critically endangered bird species endemic to New Zealand that has undergone severe population bottlenecks due to habitat fragmentation and invasive mammalian predators, reducing the entire species to just 252 individuals (as of 15/08/2022). The species is therefore highly inbred and suffers from low reproductive success (*Dussex et al., 2021*; *Lloyd and Powlesland, 1994*; *Savage et al., 2020*; *Triggs et al., 1989*; *White et al., 2015*). Kākāpō are intensively monitored by the Kākāpō Recovery Programme of the New Zealand Department of Conservation. The conservationists keep track of the home range, health, and reproductive success of each individual kākāpō by regularly handling the birds, which currently imposes financial and organizational burdens onto the conservation programme, and disturbance and stress onto the wild populations.

Here, we demonstrate that soil eDNA can reliably identify the distribution of kākāpō and other vertebrate species in a highly localized manner. We then use real-time nanopore sequencing which allows for selective sequencing based on digital genomic data (aka 'adaptive sampling'; see *Kovaka et al., 2021*; *Payne et al., 2021*) and the high-quality kākāpō reference genome (*Dussex et al., 2021*; *Guhlin et al., 2022*) to extract species-specific DNA from the soil samples. By combining the resulting long haplotypes with known genomic variation in the kākāpō population, we are able to reliably predict the presence of individuals across the kākāpō habitat. We therefore demonstrate that real-time long-read genomics can achieve individual identification in a wild species purely based on genomic material from non-invasive samples, and we showcase the utility of this approach for real-world conservation.

## Results

### Metabarcoding and species variety

We established a metabarcoding approach based on 12 S rRNA gene amplification and applied it to soil samples and negative controls (see Methods). The negative controls resulted in no sequencing output, except for one which contained some human DNA (sample 8/'control 1'; *Supplementary files 2 and 3*). As we had included a negative control for each extraction batch (n=5), we ruled out any external or cross-sample contamination in any of our samples. Across 37 soil samples taken from Whenua Hou and four samples taken from aviaries in the Dunedin Botanic Garden, we identified seven dropouts at six sites with no identifiable sequencing output (samples 6, 20, 21, 23, 41, 48, and 49; *Supplementary file 2*). These samples, however, showed good $C_q$ values (<35), suggesting that PCR inhibition was not the cause of the negative results; we therefore hypothesize that degraded DNA or the absence of any vertebrate DNA was the reason for these dropouts (*Supplementary file 2*). As we had processed two replicates per site, we were still able to report results across nearly all sample sites (except for the site at 4 m distance from feeding station 2 where both samples resulted in dropouts). After confirming that all replicates showed similar species composition (*Supplementary file 3*), we averaged the species proportions across both replicates to obtain final relative read counts per site.

Across all soil samples, we identified 21 avian and mammalian species and genera, including kākāpō (*Figure 1a*) and we found differences in relative taxa abundance between sampling locations (*Figure 1b*; *Supplementary file 3*). The kākāpō display sites contained the most kākāpō DNA, but the signal dropped quickly with increasing distance from the display sites. We also found large relative amounts of kākāpō DNA at feeding stations, but nearly none in recently abandoned nest sites, suggesting that kākāpō eDNA signals were both spatially and temporally highly resolved. We found no kākāpō DNA in the aviaries of the species' closest relatives, the *Nestor* parrots kea and kākā, but DNA of the *Nestor* parrots, humans, other exotic bird species and of invasive mammalian predators (*Figure 1b*).

### Nanopore sequencing and genomic analyses

We sequenced kākāpō-specific DNA of three soil samples with high DNA concentration, high kākāpō DNA content and a large number of long reads (samples 3, 11, and 35; *Supplementary file 1*) by using selective real-time nanopore sequencing (aka "adaptive sampling") on one GridION flow cell per sample for ~12 hr (see Materials and methods). For samples 3 and 35, technical limitations led to the production of selective and non-selective sequencing data, which we directly harnessed to compare our selective nanopore sequencing approach with 'normal' non-selective sequencing of all soil DNA contained. The non-selective nanopore sequencing approach required additional sequence filtering after sequencing to only retrieve kākāpō-specific DNA (see Materials and methods). *Table 1* summarizes the overall number of passed reads and bases (Q-score >7), and the number and percentage of reads and bases that mapped to the kākāpō reference genome. This shows that non-selective sequencing resulted in an increased relative number of mapped reads and bases (*Table 1*).

*Figure 2* shows the read distribution of all the sequencing runs. We subsequently combined the non-selective and selective sequencing to extract read-based haplotypes that were also detected in the extant kākāpō population (Materials and methods). For sample 3, we identified 30 haplotypes that completely overlapped with haplotypes present in the extant kākāpō population, for sample 11, 21 haplotypes, and for sample 35, 29 haplotypes. We subsequently calculated haplotype agreement scores that describe the percentage of overlapping haplotypes between each soil sample and each Whenua Hou kākāpō individual.

According to the haplotype agreement scores, we found that sample 3 was most similar to Moss and Sinbad (*Figure 3a*), sample 11 to Sinbad and Merv (*Figure 3b*), and sample 35 to Sinbad and Zephyr (*Figure 3c*). We combined these predictions with the extensive kākāpō metadata and found that sample 3 was taken from kākāpō Moss's display site, sample 11 from Merv's display site and sample 35 from Nora's feeding station (Materials and methods): For sample 3, we were therefore able to identify the 'correct' kākāpō of the sampled home range as the individual with the best haplotype agreement score. For sample 11, we identified the correct kākāpō individual, Merv, as the second-best hit, with Sinbad as the best hit; these two individuals together explained all haplotypes that were found in the respective soil sample. For sample 35, we again identified Sinbad as the best hit; the second-best hit was the kākāpō Zephyr, Nora's daughter.

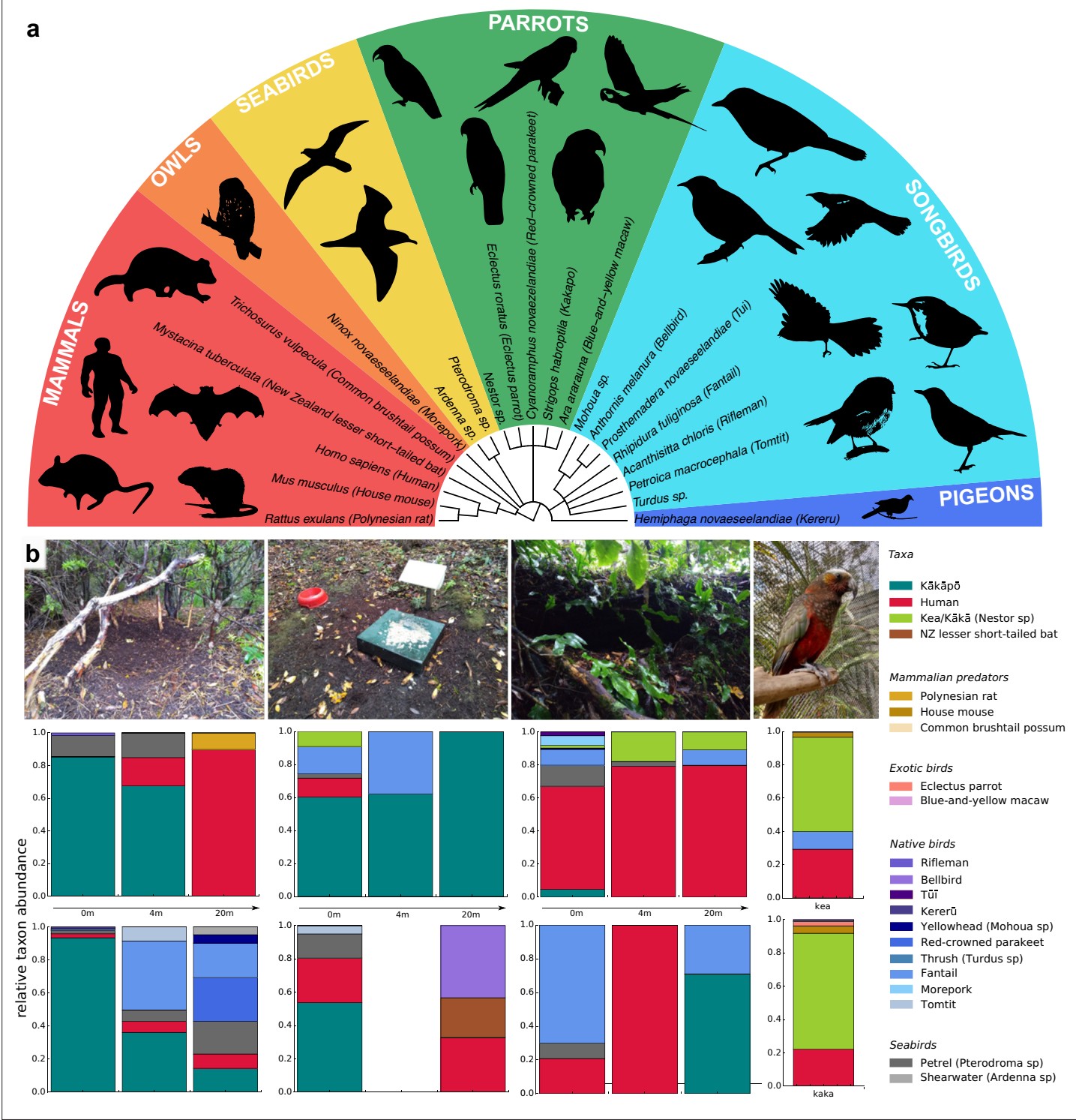

**Figure 1.** Vertebrate biodiversity in New Zealand from soil eDNA. (**a**) Cladogram of all species and genera detected by 12 S rRNA metabarcoding of soil samples from Whenua Hou and from parrot aviaries in the Dunedin Botanic Garden. (**b**) Relative taxon abundances of sampled locations averaged across replicates (*from left to right*: kākāpō display sites, feeding stations, abandoned nests, and *Nestor* parrot aviaries). Two different sites per location were sampled (*top and bottom*) at three different distances, and two aviaries of the *Nestor* species kea and kākā. For feeding station 2 (4m), both replicates resulted in dropouts.

**Table 1.** Number of passed nanopore reads and bases (Q-score >7), number of reads and bases mapping to the kākāpō reference genome, and relative amount [%] of mapped reads and bases per soil sample.
For samples 3 and 35, the results of the selective and the non-selective nanopore sequencing runs are shown.

| Sample | Run | # reads | # bases | # mapped reads | # mapped bases | % mapped reads | % mapped bases |
|---|---|---|---|---|---|---|---|
| Sample 3 | selective | 903,250 | 313,408,229 | 137 | 67,480 | 0.015 | 0.022 |
| | non-selective | 3,942,038 | 6,134,919,092 | 2,739 | 7,983,165 | 0.069 | 0.130 |
| Sample 11 | selective | 4,914,329 | 2,062,605,132 | 769 | 435,963 | 0.016 | 0.021 |
| Sample 35 | selective | 2,151,935 | 802,633,273 | 563 | 324,114 | 0.026 | 0.040 |
| | non-selective | 4,133,117 | 7,708,401,775 | 3,854 | 10,720,441 | 0.093 | 0.139 |

To investigate the omnipresent DNA signal of the kākāpō Sinbad across all samples, we used a complimentary Bayesian inference approach (Methods) to estimate mixing proportions (*Figure 3d*) and posterior means of individual assignment (*Figure 3e*) per sample. While Sinbad's signal is equally omnipresent in the mixing proportions and in the posterior means, the mixing proportion estimated all three individuals correctly when ignoring Sinbad's signal (*Figure 3e–f*). As the haplotype agreement score and our Bayesian inference approaches both predicted the presence of the kākāpō Sinbad, we analysed extensive radio transmitter and proximity sensor metadata on the movement of all individual kākāpō, which confirmed that Sinbad had indeed been close to our sampling sites three days before our sampling date (*Figure 3—figure supplement 1*).

Our maximum likelihood calculations (Materials and methods) predicted that the most likely number of contributing kākāpō individuals was three for sample 3 (MLE = $1.4 \times 10^{-6}$), two for sample 11 (MLE = $7.2 \times 10^{-4}$) and larger than five for sample 35 (MLE for six individuals = $6.4 \times 10^{-12}$). For sample 35, the MLE kept increasing with an increasing number of individuals, pointing towards several kākāpō individuals contributing DNA to this sample.

Our analysis of the background DNA of the three nanopore-sequenced soil samples found that 97% of sequencing reads were classified as of bacterial origin, and 3% as of eukaryotic origin. Most bacterial reads were assigned to the soil bacteria *Bradyrhizobium* and *Streptomyces*; other frequent taxa include typical environmental bacteria such as *Pseudomonas*, *Mycobacterium*, *Mesorhizobium*, *Burkholderia*, and *Sphingomonas*.

## Discussion

This study shows that environmental genomic material can be used to assess both species variety and within-species genomic variability in a non-invasive and efficient manner. We show that individual identification is feasible in wild populations through real-time nanopore sequencing of eDNA and subsequent long-read haplotype calling. While the prospect of non-invasive individual identification represents an important step change for the conservation of critically endangered species on its own (*Sigsgaard et al., 2020*), our approach might have additional implications for in-depth monitoring of rare and elusive species, potentially expanding the application of eDNA research from monitoring species distribution to inferring fitness-related parameters such as inbreeding, genomic diversity and adaptive potential from non-invasive genomic material.

Previous eDNA research has mostly studied the presence and distribution of species, but the potential of retrieving in-depth within-species information has been recognized for some time (*Barnes and Turner, 2016*). Our shotgun sequencing approach alleviates many challenges that are associated with traditional PCR- and amplicon-based approaches, including the risk of allelic dropout due to scarce or fragmented DNA (*Smith and Wang, 2014*) and amplification of closely related species (*Wilcox et al., 2013*), while simultaneously avoiding laborious and expensive pre-processing of DNA such as required for DNA hybridization capture and creating unbiased genomic data that can be leveraged across populations and generations despite evolutionary divergence.

We first establish that eDNA extracted from soil samples is an accurate and replicable method for monitoring a flightless bird species, the kākāpō, and for monitoring other avian and mammalian taxa. We show that less than a gram of surface soil allows for highly accurate kākāpō monitoring while detecting additional 20 species, including the elusive and threatened New Zealand lesser short-tailed

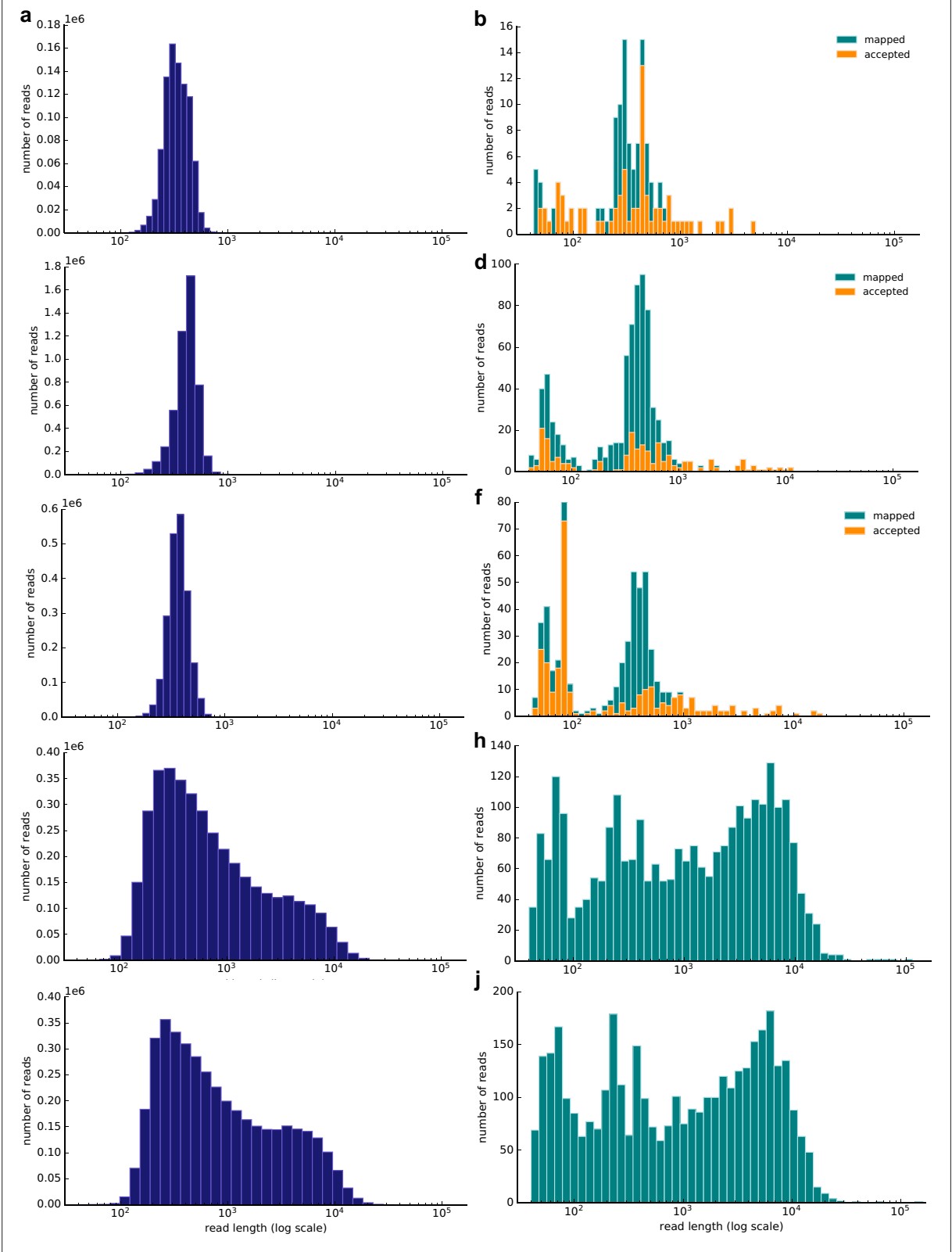

**Figure 2.** Resulting read length distribution (log10 scale) of nanopore sequencing of three exemplary soil samples (samples 3, 11 and 35; ***Supplementary file 1***). *Left*: Distribution of all passed (Q-score >7) reads; *right*: Distribution of all passed (Q-score >7) reads that map to the kākāpō reference genome using minimap2. The subset of mapped reads that have been accepted by selective sequencing (not 'unblock' reads; Methods) is highlighted in orange. The selective sequencing results are shown by (**a**) and (**b**) (Sample 3), (**c**) and (**d**) (Sample 11), (**e**) and (**f**) (Sample 35). The non-

*Figure 2 continued on next page*

selective nanopore sequencing data is shown by (**g**) and (**h**) (Sample 3) and (**i**) and (**j**) (Sample 35). The selective runs result in many reads of ~500 bp length, which is the average sequencing length at which reads are long enough to be taken a decision upon and to be rejected.

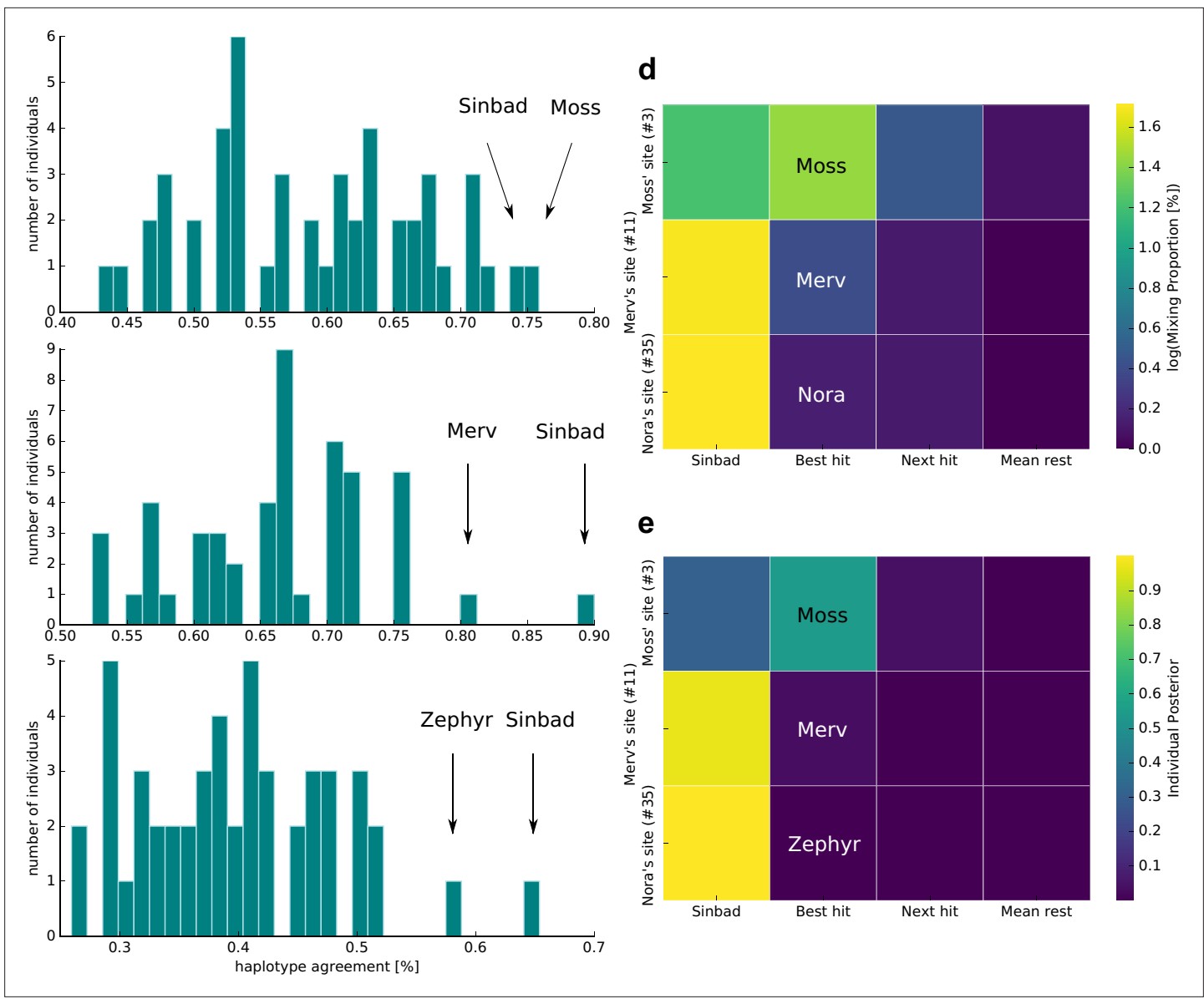

**Figure 3.** Individual identification from soil nanopore sequencing data. (**a–c**) Distribution of haplotype agreement scores between all Whenua Hou kākāpō and (**a**) soil sample 3 (Moss' display site), (**b**) soil sample 11 (Merv's display site), and (**c**) soil sample 35 (Nora's feeding station). (**d**) Mixing proportions [%; log10 scale] and (**e**) posterior means of individual assignment per sample (*y-axis*) assessed through Bayesian inference of individual assignments (see Materials and methods). The heatmaps show Sinbad's omnipresent signal in the first column, the best hit when disregarding Sinbad in the second column, the second-best hit in the third column, and the mean values of all remaining Whenua Hou kākāpō in the last column.

The online version of this article includes the following figure supplement(s) for figure 3:

**Figure supplement 1.** Stamen terrain map of the sampling sites of the three soil samples 3 (Moss' site), 11 (Merv's site), and 35 (Nora's site), and of the radio transmitter signal receiver that recorded Sinbad's presence far away from his home range ('Sinbad's bowl') and in the middle of the sampling sites, on February 24, 2019, three days before our sampling efforts.

bat (*Mystacina tuberculata*), and invasive mouse and possum species. We importantly detected a few reads of the invasive Polynesian rat (*Rattus exulans*) on Whenua Hou, which serves as a predator-free sanctuary for the surviving kākāpō population. As we only found very weak evidence in only one sample, we however postulate that the rat genomic material might have been transported to the island via avian predators. Alternatively, contamination could have happened in the laboratory which handles ancient *R. exulans* samples. The application of soil eDNA research can therefore make an essential contribution to conservation by enabling efficient detection of both endangered and invasive predators, obviating other labor- and cost-intensive invasive methods that are currently being employed in New Zealand and around the world. Our soil eDNA approach has a high spatial and temporal resolution, as shown by the rapid drop in signal with increasing distance from kākāpō hotspots and by the scarcity of kākāpō DNA in recently abandoned nests. We show that our approach can distinguish DNA from kākāpō from its closest relatives, the *Nestor* parrots kea and kākā: We, as expected, did not find any kākāpō signal in the artificial *Nestor* aviaries, but evidence of *Nestor*, human, exotic birds, and mammalian pest DNA.

We then show that real-time nanopore sequencing can be leveraged for non-invasive individual identification in wild populations. To overcome problems associated with increased sequencing error rates of nanopore sequencing, we use the long nanopore reads to create robust haplotypes. We importantly observe that the low proportion of target DNA in our soil samples (<0.1%) makes standard 'non-selective" nanopore sequencing even more efficient than selective nanopore sequencing (aka 'adaptive sampling') at producing species-specific sequencing reads. When the target DNA is less than 0.1% in the selective sequencing approach, more time will be spent on read-unblocking than sequencing target DNA (internal communications with ONT). We therefore recommend determining the target DNA content in an exploratory sequencing run to evaluate the potential of selective sequencing. We, however, anticipate that selective nanopore sequencing will rapidly increase in efficiency, resulting in reduced pore-clogging and potentially faster decision-making with less sequencing efforts spent on read-unblocking. Selective nanopore sequencing further brings the prospect of targeting finer scales, such as selecting specific chromosomes or genomic regions that are highly representative of a species' genome-wide diversity. Standard non-selective nanopore sequencing can, on the other hand, be advantageous since it allows for within-species assessments across multiple taxa, combining species detection with within-species monitoring. We also show that nanopore sequencing can provide a more holistic view of an ecosystem by simultaneously assessing its microbiome by successfully ascertaining the soil's characteristic bacterial composition.

We achieved individual identification through two complementary approaches, haplotype agreement scoring and Bayesian inference. As our haplotype agreement scores require an exact overlap between soil- and population-based haplotypes to stringently account for nanopore sequencing errors, the number of remaining haplotypes is sparse (ranging from 21 to 30). We anticipate that this approach can be more lenient in the future given the increasing accuracy of nanopore sequencing of >99%. A larger number of haplotypes might then allow us to cover a larger proportion of the genome and to discern family relationships more accurately. This could resolve individual identification in highly inbred populations, which has been limited in our current approach where we were not able to discern the genomic signal of the kākāpō Nora and her offspring. We, however, anticipate that the accuracy of our presented approach is already sufficient for delineating families and subpopulations in a wild species, allowing for in-depth population-based conservation management.

We also leveraged Bayesian inference approaches for conditional genetic stock identification to infer contributions of kākāpō individuals to the soil sample. This approach together with our maximum likelihood estimations of the number of contributing individuals confirmed the unexpected detection of the kākāpō Sinbad in our samples. Leveraging extensive metadata of kākāpō whereabouts, we were able to show that Sinbad had indeed visited a location close to our sampling sites three days before data collection. This shows that our approach can accurately describe mixtures of individuals, which will be essential for monitoring non-territorial and migrating species.

We here show that nanopore sequencing can enable real-time in-depth monitoring of wild populations, both on the level of species variety and individual genomic variability. This further indicates that it might be feasible to assess the genetic health and adaptive potential of wild populations in a completely non-invasive manner. Our approach will, as a tangible example, directly assist the kākāpō conservationists in monitoring individuals in an efficient and non-invasive manner, and in detecting

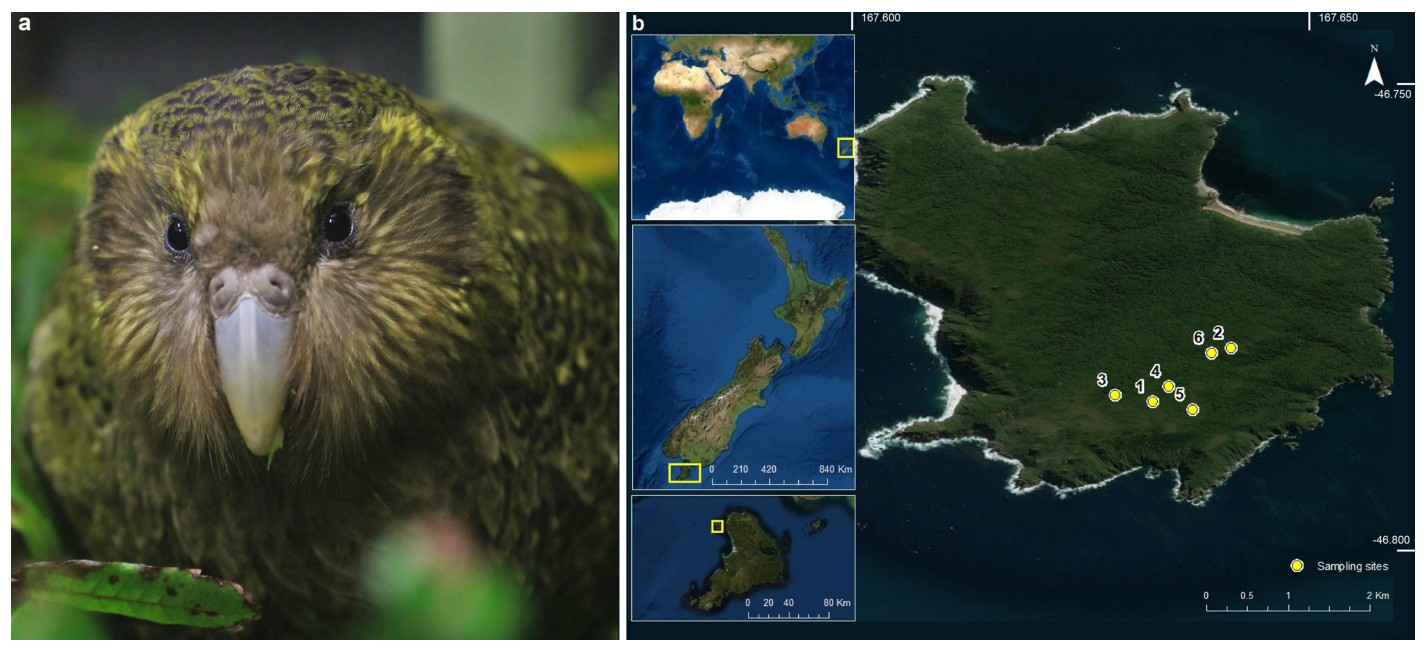

**Figure 4.** The critically endangered kakapo and its New Zealand habitat. (**a**) A kākāpō (*picture credit*: Lydia Uddstrom). (**b**) Map enhancement of sampling locations on Whenua Hou, New Zealand (*service layer credit*: Esri, Maxar, GeoEye, Earthstar, Geographics, CNES/Airbus DS, USDA, USGS, AeroGRID, IGN, and the GIS User Community).

potentially remnant populations in the wild. Even more importantly, we show that the integrated application of eDNA to detect endangered and invasive species and to monitor individuals and subpopulations in endangered populations has the potential to substantially aid universal conservation management around the globe.

## Materials and methods
### Sample acquisition

Soil sampling was performed on Whenua Hou, New Zealand, the island with the largest kākāpō population (*Figure 4a*), on February 27, 2019. We sampled sites of interest, including male display sites (shallow bowls in the ground that are frequented by males every night during breeding seasons), recently abandoned nests (~30 days), and supplementary feeding stations. At each site, sampling time, location, and environmental observations were recorded (*Figure 4b*). Per site, a new set of nitrile gloves was used for sample collection, debris and leaf litter were removed and a finger-full of soil (~5–10 g) from the surface was put into a small sterile plastic bag; two replicates were taken at each site and subsequently stored in a medium-sized bag to avoid cross-contamination across replicates. Per site, samples and respective replicates were additionally taken at distances of 4 and 20 m. All samples per site were stored in a large plastic bag to avoid cross-contamination across sites, and frozen at –20 °C as soon as possible (at the latest after five hours). Altogether, 37 samples were taken at six sites (*Supplementary file 1*; *Figure 4b*).

We additionally sampled soil in aviaries of the kākāpō's two closest relative species, the kea (*Nestor notabilis*) and kākā (*Nestor meridionalis*). We sampled two aviaries per species in the Dunedin Botanic Garden, Dunedin, New Zealand, on September 18, 2020.

### DNA extraction

DNA was extracted from the Whenua Hou (n=37) and Botanic Garden (n=4) soil samples using Qiagen's DNeasy PowerSoil Pro kit following the manufacturer's recommendations for extraction of genomic DNA from average-wet soil (recommended amount of 250 mg; for exact amounts of soil, see *Supplementary file 1*). Extractions were performed in a designated PCR-free hood which was cleaned with bleach and deionized water, followed by UV exposure for 30 min. A cleaned benchtop

(wiped with bleach and deionized water) and a new set of nitrile gloves were used for each sample during the initial extraction step (weighing and placing samples into new PowerSoil tubes). In addition, extraction negative controls (deionized water) were included in every extraction run (n=5) to ensure no contamination was introduced during the extraction process (*Supplementary file 1*). The extracted DNA concentration was measured using the Qubit 4 Fluorometer (Thermo Fisher Scientific) (*Supplementary file 1*) and stored at –20 °C. We additionally assessed the DNA fragment length distribution of several DNA extracts using the QIAxcel gDNA High Sensitivity protocol (QX DNA Size Marker of 250 bp–8 kb and QX Alignment Marker of 15 bp/10 kb).

## Metabarcoding and amplicon sequencing

DNA quality/quantity analysis, adapter-fusion, indexing and amplification were carried out in single-step quantitative PCR reactions on an Applied Biosystems QuantStudio 1 qPCR instrument. DNA extracts were PCR-amplified using the "RV" fusion-tag mitochondrial 12 S rRNA-V5 ecoPrimers for the detection of bird, mammalian and fish species (RV forward primer: 3`-<u>AATGATACGGCGACCACCGA GATCTACAC</u>TGACGACATGGTTCTACAXXXXXXXXX**GACGTTAGATACCCCACTATGC**-5`; RV reverse primer: 5`-<u>CAAGCAGAAGACGGCATACGAGAT</u>XXXXXXXX **TAGAACAGGCTCCTCTAG**-3`; adapted from *Riaz et al., 2011*; shown with Illumina P5 and P7 adapter sequences underlined, Illumina TruSeq sequencing primer-binding site unmarked, 8 bp unique index tags as X strings and locus-specific primers in bold). All 8 bp index tags differed from each other by at least 3 bp. Each reaction contained 5 µl SensiFAST 1 x LoRox SYBR Mix (Bioline), 0.25 µl forward primer (10 µM), 0.25 µl reverse primer (10 µM), 0.5 µl BSA (10 mg ml$^{-1}$, Sigma Aldrich), 2 µl deionised water and 2 µl template DNA. qPCR cycling conditions included an initial denaturation of 3 min at 95 °C, followed by 40 cycles of 5 s at 95 °C, 10 s at 52 °C, and 15 s at 72 °C. DNA quality and quantity were confirmed by assessing that a sigmoidal log-amplification curve was visible at a $C_q$ value of <35. A negative control reaction containing 2 µl of deionised water in place of the template DNA was included with each run.

Sequencing libraries were pooled at approximately equimolar concentration using the final normalized ΔRn fluorescence values as a guide and cleaned and double-end size selected using AMPure XP magnetic beads (0.9 x and 1.2 x for lower and upper size bounds, respectively). The final pooled library concentration was determined using a Qubit 4 Fluorometer (Thermo Fisher Scientific) and the concentration was adjusted to 50 pM in sterile DNAse/RNAse-free water. The library was then loaded onto an iSeq i1 V2 reagent cartridge with a 300-cycle flow cell (Illumina) with 5% Phi X and run for 290 cycles in a single direction on an Illumina iSeq 100 (see *Supplementary file 2* for number of sequencing reads).

## Amplicon sequence variant generation and taxonomic assignment

The iSeq output FASTQ files were de-multiplexed using the R programming language (*R Development Core Team, 2021*), using the insect package v1.4.0 (*Wilkinson et al., 2018*) trimmed sequences were filtered to produce a table of exact ASVs using DADA2 (*Callahan et al., 2016*). ASVs were identified to the lowest possible taxonomic rank using the following process: (1) ASVs were exact-matched against a New Zealand-specific database of previously detected eDNA sequences curated by Wilderlab; (2) remaining (i.e., non-matched) ASVs were exact-matched against a larger local reference sequence database compiled of trimmed 12 S rRNA sequences from GenBank (*Benson et al., 2009*) and BOLD (*Ratnasingham and Hebert, 2007*) matching ASVs were assigned at the lowest common ancestor level (LCA; assigned to genus level if matched with 100% identity to more than one species, or to family level if matched to more than one genus); (3) remaining ASVs that were >50 bp in length were matched with single indel/substitution tolerance against the same GenBank/BOLD reference database and matching ASVs were assigned at LCA level; and, finally, (4) remaining ASVs were queried against the local GenBank/BOLD reference database using the SINTAX classification algorithm (*Edgar, 2016*) with a minimum conservative assignment threshold of 0.99 and genus level as maximum taxonomic resolution (*Supplementary file 2*). We subsequently restricted the taxonomic assignments to the species and genus level to only consider highly resolved ASVs (*Supplementary file 3*); we hereby included the genus level since DNA sequence databases are incomplete with respect to New Zealand's fauna and therefore often do not allow taxonomic assignment to the species level. We further removed samples from *Supplementary file 3* that produced either no reads or only unintelligible reads with no species or genus taxonomic classifications.

**Table 2.** Details of the three soil samples subjected to nanopore sequencing.
DNA concentration after bead clean-up [ng/ul]; volume used as input for library preparation [ul] to achieve a DNA input amount of 1 μg per library preparation; amount of DNA in the final library [ng] used as input for sequencing; number of active pores per nanopore flow cell; and metadata of each sample.

| Sample number | Concentration after clean-up [ng/ul] | Volume library preparation [ul] | Final amount of DNA in library [ng] | # pores | Metadata |
| --- | --- | --- | --- | --- | --- |
| 3 | 91.2 | 12.3 | 584 | 1547 | Display site of kākāpō individual Moss |
| 11 | 202.0 | 5.0 | 257 | 1374 | Display site of kākāpō individual Merv |
| 35 | 119.0 | 12.0 | 615 | 1712 | Feeding station of kākāpō individual Nora |

## Nanopore sequencing

Based on the 12 S rRNA amplicon analysis, we identified three samples (samples 3, 11 and 35) with high DNA concentrations (>200 μg/ml; *Supplementary file 1*), many kākāpō-assigned 12 S rRNA reads (>1500; *Supplementary file 3*) and a strong peak at the maximum read length (at ~10 kbp; upper limit of the QIAxcel gDNA High Sensitivity protocol). We subsequently prepared these samples for nanopore sequencing. Briefly, we prepared the sequencing libraries using the SQK-LSK109 protocol, following the manufacturer's recommendations. We added a bead-cleanup step before library preparation, using a 1:1 mixture of deionized water and freshly prepared 80% ethanol, to remove any small DNA fragments. We then used 1 μg of DNA as input for library preparation and diluted the final library in 15 μl elution buffer (*Table 2*). We extended the incubation of DNA repair and end-preparation to 30 min at 20 °C (followed by the standard 5 min at 65 °C), used the kit's Short Fragment Buffer, and incubated the library for 10 min at 37 °C at the end of library preparation to improve the recovery of long reads. We then loaded one library per sample onto an R9.4.1 flow cell and ran them for approximately 12 hr on a GridION Mk1, using the FAST basecalling mode and the high-quality kākāpō reference genome (NCBI taxonomy ID: 2489341) as digital target sequence template (*Table 2*).

## Nanopore sequencing data processing

We used Guppy v3.2 (*Wick et al., 2019*) for high accuracy (HAC) basecalling and adapter trimming of all passed output reads (across all selective sequencing decisions, including 'unblock' for rejected reads, 'no_decision' for reads that were too short for a decision to be taken, and 'stop_receiving' for accepted reads). We then used Nanofilt v2.6 (*De Coster et al., 2018*) to filter all reads for quality (Q-score >7) and aligned all reads to the kākāpō reference genome using minimap2 v2.17 (*Li, 2018*). We included all reads since some of the rejected and undecided reads aligned to the reference genome using minimap2 but were not included as accepted reads, mostly due to their short length. We then used SAMtools v1.10 (*Li et al., 2009*) to transform the resulting sam files to sorted bam files, filter the bam files for mapped reads, index them and count the number of mapped reads.

We used Medaka v1.2.5 (*nanoporetech, 2019*) to call variants against the reference genome using *medaka_variant*; *medaka_variant* intrinsically uses WhatsHap (*Martin et al., 2016*) to estimate the underlying haplotypes per genomic site. We used these haplotype probabilities and the *medaka snp* command to create a gvcf variant file. We then compared the resulting vcf (which, as opposed to the gvcf file also contains indels) and gvcf files with an existing population-wide genomic variant callset using the Python package PySAM v0.15.3 (*pysam-developers, 2022*). Briefly, the existing population variant callset was produced by the Kākāpō125+consortium by applying DeepVariant (*Poplin et al., 2018*) to the genomic dataset of nearly the entire kākāpō population (n=169 out of 171 alive kākāpō as of 31/12/2018) and by filtering the resulting high-quality variant set for genotype missingness of <20% and a minor allele frequency >1% (resulting in 1,612,477 variants; *Guhlin et al., 2022*).

To account for sequencing errors due to potentially degraded DNA and the increased sequencing error rate inherent to nanopore's R9.4.1 flow cell chemistry (estimated at ~8% by *Urban et al., 2021*), we used customized Python 3.5.2 scripts and only retained the variants that were identical in location and alleles in both, the soil and population variant callsets. We then retrieved the soil haplotypes as estimated by Medaka and assigned the variants to haplotypes. Again, we only retained those haplotypes

that matched between the soil and population variant callsets. Based on these identical haplotypes, we calculated haplotype agreement scores between each soil sample and every individual in the population variant callset. We additionally used the R package rubias v0.3.2 (*Moran and Anderson, 2019*) to apply Markov Chain Monte Carlo (MCMC) methodology with a uniform prior distribution for estimating mixture proportions and individual posterior probabilities of assignment through MCMC iterations conditional on the reference allele frequencies (2000 iterations; burn-in of 100).

We performed contributor analyses to estimate the most likely number of kākāpō individuals contributing to each sample. We used a combinatorial maximum likelihood analysis based on our haplotypes, with the likelihood of each combination of individuals being calculated as the product of per-haplotype probability. The per-haplotype probability was calculated as the relative number of individuals that matched the soil haplotype. To account for missing values, we mean-imputed the missing values across individuals on a per-haplotype basis. We calculated the maximum likelihood estimate (MLE) of $n=1$ up to $n=6$ individuals contributing to each of the soil samples.

We finally assessed the taxonomic origin of the background DNA produced by nanopore sequencing approach (i.e., reads classified as 'unblock' or 'no_decision') using ONT's cloud-based EPI2ME's What's in my Pot (WIMP) (*Juul et al., 2015*) platform.

All plots were produced in Python, using Matplotlib v1.5.3.

## Acknowledgements

This research was funded by grants to LU from Birds New Zealand, the Department of Conservation, the University of Otago, the Alexander von Humboldt Foundation and Revive & Restore. The authors thank the Kākāpō Recovery Programme for the fantastic collaboration, and Te Rūnanga o Ngāi Tahu as kaitiaki of this taonga species. Special thanks go to the Whenua Hou committee, who allowed us to participate in their hui, and to Tane Davis. The development of the kākāpō genomic data was supported by Genomics Aotearoa and by Kākāpō125+. The authors also wish to acknowledge NeSI (New Zealand eScience Infrastructure; especially Dinindu Senanayake), Miles Benton, Eddy Dowle, Alana Alexander, Joanne Gillum, Tim Moser, Tim Hore, Otago Genomics (especially Aaron Jeffs), Patricia Fuentes-Cross, and the Dunedin Botanic Garden (especially Alisha Sheriff).

## Additional information

### Competing interests

Megan Shaffer, Shaun P Wilkinson: is affiliated with Wilderlab NZ Ltd. The author has no other competing interests to declare. The other authors declare that no competing interests exist.

### Funding

| Funder | Grant reference number | Author |
| --- | --- | --- |
| Alexander von Humboldt Foundation | | Lara Urban |
| Department of Conservation, New Zealand | | Lara Urban |

The funders had no role in study design, data collection and interpretation, or the decision to submit the work for publication.

### Author contributions

Lara Urban, Conceptualization, Resources, Data curation, Formal analysis, Supervision, Funding acquisition, Investigation, Visualization, Methodology, Writing – original draft, Project administration, Writing – review and editing; Allison K Miller, Resources, Visualization, Writing – review and editing; Daryl Eason, Deidre Vercoe, Data curation, Methodology; Megan Shaffer, Shaun P Wilkinson, Resources, Data curation, Formal analysis; Gert-Jan Jeunen, Resources; Neil J Gemmell, Project administration, Writing – review and editing; Andrew Digby, Conceptualization, Data curation, Visualization, Methodology, Project administration, Writing – review and editing

## Author ORCIDs
Lara Urban http://orcid.org/0000-0002-5445-9314
Allison K Miller http://orcid.org/0000-0002-5334-2771
Neil J Gemmell https://orcid.org/0000-0003-0671-3637

Reviewer #1 (Public Review): https://doi.org/10.7554/eLife.84553.2.sa1
Reviewer #2 (Public Review): https://doi.org/10.7554/eLife.84553.2.sa2
Author Response: https://doi.org/10.7554/eLife.84553.2.sa3

## Additional files

### Supplementary files
• Supplementary file 1. Information about soil samples including details about sampling and DNA extraction. Sample ID, Sample Type, Distance from the sampling site [m], Amount of soil sampled [ug], DNA concentration after DNA extraction [ug/ml], Date of extraction (i.e. extraction batch), and Sample label, which is used throughout the manuscript.

• Supplementary file 2. Number of 12 S rRNA reads per soil sample. All reads, reads that could be classified, reads that map to any Metazoan taxa, and reads that mapped to the taxonomic species level.

• Supplementary file 3. Number of 12 S rRNA reads mapping to each detected species and genus (rows) across all soil samples (columns). For each taxon, the taxonomic rank, scientific and common name, and the NCBI taxonomic ID are indicated.

• MDAR checklist

### Data availability
The raw data can be accessed at NCBI (BioProject ID PRJNA806467 for metabarcoding data; ID PRJNA812072 for nanopore sequencing data). Custom Python 3.5.2 code and R scripts are available via GitHub (copy archived at *Urban, 2022*). The kākāpō; population genomic dataset and respective genomic variant callset are available via an application form at the Aotearoa Genomic Data Repository as per the Global Indigenous Data Alliance guidelines. Access to the data is controlled by a data committee composed of the Department of Conservation and Te Rūnanga o Ngāi Tahu.

The following datasets were generated:

| Author(s) | Year | Dataset title | Dataset URL | Database and Identifier |
|---|---|---|---|---|
| Urban L | 2022 | Genomic monitoring of the critically endangered kakapo by real-time nanopore sequencing of environmental DNA | http://www.ncbi.nlm.nih.gov/bioproject/PRJNA812072 | NCBI BioProject, PRJNA812072 |
| Wilkinson S | 2022 | Genomic monitoring of the critically endangered kakapo using targeted nanopore sequencing of environmental DNA | http://www.ncbi.nlm.nih.gov/bioproject/PRJNA806467 | NCBI BioProject, PRJNA806467 |

The following previously published dataset was used:

| Author(s) | Year | Dataset title | Dataset URL | Database and Identifier |
|---|---|---|---|---|
| Digby A | 2022 | Kakapo125+ genome sequencing | https://repo.data.nesi.org.nz/discovery/TAONGA-KAKAPO/ | Aotearoa Genomic Data Repository, kākāpō |

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
