## [Editor Report · eLife assessment]

This work presents **important** findings regarding the use of soil environmental DNA for non-invasive monitoring of the endangered kākāpō parrot population in New Zealand. The approach based on sequence analysis is **convincing** but comparisons to established methods are lacking. The tools presented in this study are innovative and will be relevant to those working with environmental DNA and the conservation of biodiversity.

---

## [Referee Report · Reviewer #1 (Public Review)]

The fields of ancient and environmental DNA have many similarities. Practitioners are constantly tinkering with methods to extract as much information from biological samples as possible. Both fields of research also have to deal with the fact that only a tiny fraction of the DNA is 'on target' and that the background DNA (largely bacterial) is often immense.

In this research Urban et al tackle the question of individual identification of a flightless New Zealand parrot (the kakapo) using shotgun eDNA (from soil) within a study system where reference genomes exist for most of the animals within a population. Most eDNA studies stay in the relative safety of metabarcoding (typically on mitochondrial DNA) thus Urban et al are breaking new ground.

In this small-scale (and highly controlled) study, Urban et al. use shotgun eDNA from a gram of soil and then match kakapo reads to reference genomes. Using some innovative Bayesian inference the researchers are able to identify individuals within the populations.

There are a number of innovations in this study that have relevance to the conservation sector. The idea that we can identify individuals in a population in a non-invasive manner is an exciting prospect. It immediately conjures up the possibility of genetic mark-recapture applications. In the case of highly endangered populations, the work shows the value of building reference genomes for the whole population.

At its core, this is a proof-of-principal study that arguably leaves the reader with more questions than answers. I was left wondering (i) why didn't nanopore's adaptive sampling function enrich targets? (ii) how would short-read platforms compare (iii) could genomic signatures of other taxa (e.g. bats) identified by metabarcoding be detected in shotgun data? And (iv) is sediment the best substrate for this work?

Sedimentary DNA methods have been around for ~20 years and it is exciting to see the field continue to innovate. The speed and portability of nanopore devices may, with time, see real-time genotyping become a reality in conservation biology. I welcome these innovations as, on the global stage, we need all the tools we can get to battle the biodiversity crisis.

---

## [Referee Report · Reviewer #2 (Public Review)]

This study uses DNA metabarcoding to identify vertebrates and kākāpō DNA in soils from sites where they are known to occur and from control sites housing related birds. The authors then attempt to identify individual kākāpō birds that have contributed DNA into just three samples with high kākāpō DNA content. For this, they use Oxford Nanopore adaptive sequencing, haplotype identification, and two statistical approaches to determine the number of individuals that contributed to a sample and which specific individuals contributed. This study builds on recent developments in the field that move eDNA into population genomics and individual surveillance.

The manuscript introduction does a satisfactory job of contextualizing the need for this study and the state of the field. It does not detail the challenges of applying adaptive ONT to eDNA samples and the kinds of choices such as selective assays available. I think the authors are using confusing language in the abstract and throughout that is not clear enough to be useful to a reader community that is interested in adopting ONT but not already using it.

As for the methods chosen for this study, I found it peculiar that the authors did not use qPCR specific to kākāpō to estimate the relative proportion of kākāpō eDNA to other vertebrate DNA in the total sample. A fair comparison of methods would make this study more useful to guide the field forward. qPCR should be more sensitive than metabarcoding and is the standard approach for the relative abundance that the terrestrial eDNA community uses for targeted studies.

There is a lot of work done in this study that would be useful to the eDNA community if it were presented clearly. Paragraphs are written often without topic sentences, headings are vague, specific objectives are not clearly outlined, and too many questions remain about why certain approaches were used. For example, there is a selective and non-selective approach used for ONT sequencing. In some places, is not clear what exactly the authors did, and it's not clear why the non-selective approach was preferred by the authors (as they describe in the discussion). The ONT portion of the methods seems written out of order and with frivolous choices about what details to include and omit. No mention of the pore destruction of selective/adaptive sequencing is described, so this study creates hyperbole about the promise of ONT unblocking pores for future research. There are drawbacks! Further, there surely is going to be a lot of interest in the statistical approaches to infer individuals and the number of individuals that shed DNA into a sample but this is not clearly explained. An effort to improve the writing quality throughout is needed prior to publication.

The study fails to describe the scale of the sites and how they are managed. As such, we cannot assess the distance from the site and why kākāpō DNA was found at an abandoned nest site. Maybe it was clear but the names of the sites are inconsistent throughout the ms, and there are assumptions that readers know about this field setting already, which is not a good assumption to make.

The discussion cites nobody and does not put the results back into the broader context of where the science is today. It is a weak discussion that just reiterates the results, but then boasts about the significance of the results when those results referred to were insufficiently described in the manuscript.

Altogether, I think this study has potential if the paper can be improved in clarity and quality. The science is solid and the topic is of great interest to a broad community.

---

## [Author Response]

Dear eLife Editorial Board, dear reviewers, dear readers,

We very much thank the eLife editors and reviewers for their overall very positive review and encouraging assessment of our manuscript, and for highlighting our study’s innovation and relevance for using genomic approaches for the conservation of biodiversity.

We very much thank the reviewers for pointing out parts of the manuscript that could be described more clearly or in more detail to make the study fully reproducible, and have therefore rewritten parts of the manuscript. We importantly follow reviewer 1’s specific recommendation to focus the main text on clearly understandable results, and therefore now only showcase the application of selective nanopore sequencing (aka adaptive sampling) to one soil sample, which we hope will make the flow of the manuscript easier to understand.

We further agree that parts of the study could have been conducted more extensively (e.g. include more samples and thereby showcase the broad applicability of the approach), which was unfortunately not feasible since I as the lead author left New Zealand to take up another position abroad. We are, however, following up on this work with another controlled large-scale study.

We further agree that both qPCR and metabarcoding have their advantages and disadvantages. Metabarcoding approaches, however, importantly deliver more information about the biodiversity of a location than just the presence of a single species; this, in our case, includes other endangered species and evidence of kākāpō predators. We further show that the chosen marker gene region (12S rRNA) is species-specific enough to distinguish kākāpō from its two closest relatives. While qPCR has been shown to be more sensitive for some species, the difference is often minimal (see e.g., Harper et al., Ecol Evol. 2018 Jun; 8(12): 6330–6341), and for some species has been shown to be equally sensitive (Schneider et al., PLoS ONE 2016, 11, e0162493). qPCR approaches further require the careful design of species-specific primers, and herewith the access to samples and DNA of the target species and of closely related species – all of which are not necessarily at hand, especially not for conservationists who want to use these approaches regularly in the future, and in countries like New Zealand where genomic work with material from any “treasured” species has to be approved in a long and detailed process according to national regulations and the Nagoya Protocol. Given all these reasons, and the general good performance of our metabarcoding approach (also in detecting our species of interest), we do not see the necessity of applying a qPCR approach in this study.

To avoid any confusion, we now also describe the samplings sites in more detail and use their labels consistently throughout the manuscript. Briefly, the sites were always sampled directly at the site, and at 4m and 20m distance, and all in replicates, as described in detail in the manuscript. Specifically, the “abandoned nests” had only been abandoned ~30 days before sampling, as described in the Methods, and this is why kākāpō DNA is still present.

We further thank reviewer 2 for suggesting to discuss the impact of selective nanopore sequencing on pore efficiency in more depth, and added a respective sentence to the Discussion. We in general added more references and the broader scientific context to the Discussion.

Thank you again for this very helpful review of our work.

With best regards,

Lara Urban